# ATG5 is instrumental in the transition from autophagy to apoptosis during the degeneration of tick salivary glands

Yanan Wang[1], Houshuang Zhang[1], Li Luo[1], Yongzhi Zhou[1], Jie Cao[1], Xuenan Xuan[2], Hiroshi Suzuki[2], Jinlin Zhou[1]*

**1** Key Laboratory of Animal Parasitology of Ministry of Agriculture, Shanghai Veterinary Research Institute, Chinese Academy of Agricultural Sciences, Shanghai, China, **2** National Research Center for Protozoan Diseases, Obihiro University of Agriculture and Veterinary Medicine, Obihiro, Hokkaido, Japan

* jinlinzhou@shvri.ac.cn

**Data Availability Statement:** All relevant data are within the manuscript and its Supporting information files.

## Abstract

Female tick salivary glands undergo rapid degeneration several days post engorgement. This degeneration may be caused by the increased concentration of ecdysone in the hemolymph during the fast feeding period and both autophagy and apoptosis occur. In this work, we first proved autophagy-related gene (ATG) and caspase gene expression peaks during degeneration of the tick salivary glands. We explored the regulatory role of *Rhipicephalus haemaphysaloides* autophagy-related 5 (RhATG5) in the degeneration of tick salivary glands. During the fast feeding phase, RhATG5 was cleaved and both calcium concentration and the transcription of Rhcalpains increased in the salivary glands. Recombinant RhATG5 was cleaved by μ-calpain only in the presence of calcium; the mutant RhATG5$^{191-199\Delta}$ was not cleaved. Treatment with 20-hydroxyecdysone (20E) led to programmed cell death in the salivary glands of unfed ticks *in vitro*, RhATG8-phosphatidylethanolamine (PE) was upregulated in ticks treated with low concentration of 20E. Conversely, RhATG8-PE decreased and Rhcaspase-7 increased in ticks treated with a high concentration of 20E and transformed autophagy to apoptosis. High concentrations of 20E led to the cleavage of RhATG5. Calcium concentration and expression of Rhcalpains were also upregulated in the tick salivary glands. RNA interference (RNAi) of RhATG5 *in vitro* inhibited both autophagy and apoptosis of the tick salivary glands. RNAi of RhATG5 *in vivo* significantly inhibited the normal feeding process. These results demonstrated that high concentrations of 20E led to the cleavage of RhATG5 by increasing the concentration of calcium and stimulated the transition from autophagy to apoptosis.

## Author summary

Ticks are well-known pathogen vectors which transmitted virus, bacterial and protozoan. They are considered to be second only to mosquitoes as global vectors of human diseases. Most tick-borne pathogens (TBPs) are transmitted to hosts through tick bites assisted by saliva. Control of ticks has been achieved primarily by the application of acaricides, a

**Funding:** This work was supported by a grant No. 31572512 from the National Natural Science Foundation of China (NSFC) for Dr.Jinlin Zhou. The funders had no role in study design, data collection and analysis, decision to publish, or preparation of the manuscript.

**Competing interests:** The authors have declared that no competing interests exist.

method that has drawbacks such as environmental contamination and selection of pesticide-resistant ticks. Understanding the tick physiological characteristics is the key step for this objective; however, there are knowledge gap remained in tick physiology. Tick salivary glands rapidly degenerate and disappear within 4 days post engorgement. In this research, we are focused on tick salivary glands rapidly degeneration within 4 days post engorgement, and made several highlights findings: The first work demonstrated that 20E promotes both autophagy and apoptosis during tick salivary gland degeneration; RhATG5 is the first reported ATG5 homologue in ticks; RhATG5 play an important role in both autophagy and apoptosis during the degeneration of tick salivary glands.

## Introduction

Ticks, which are well-known pathogen transmission vectors, are long-lived, reproductively prolific, and can utilize a variety of hosts [1–3]. Ticks also may remain attached to the host, feeding on host blood, for long periods [4]. Most TBPs are transmitted to hosts through tick bites assisted by saliva [3,5–8]. During the blood-sucking process, tick salivary glands secrete a variety of biologically-active molecules, which reduce host pain and itching, inhibit blood coagulation, disturb the innate and adaptive immune responses, and dilate the blood vessels [7–14]. These molecules contribute to the spread, survival, and reproduction of TBPs [15].

Tick salivary glands rapidly degenerate and disappear within 4 days post engorgement [7,16]. During the degeneration process, many autophagic vacuoles appear in salivary gland cells and tick secretion ability decreases up to 90% [17]. Salivary gland degeneration begins during the initial period of fast feeding, and may primarily be caused by the increase in ecdysone in the hemolymph [18,19]. To verify the association between ecdysone and salivary gland degradation, 20E was added to an *in vitro* culture of salivary glands that had been extracted post-feeding, the wet weight and secretion abilities of the salivary glands decreased significantly after incubation with 20E [17]. However, the mechanisms underlying these effects remain unclear.

Autophagy involves two evolutionarily conserved ubiquitin-like conjugation systems, as follows: the ATG12-ATG5 (participate in the formation of the autophagosome membrane); and ATG8 can be divided by ATG4B and then connected to PE to form ATG8-PE, which is marked in the double-membrane of autophagosomes and as an indicator of autophagy [20–22]. Apoptosis is characterized by DNA fragmentation and caspase activation; therefore, caspase activity usually used to display apoptosis [23]. Caspase-3 is an executor that indicates apoptosis [24]. The upregulation of cleaved-caspase-3 level indicates apoptosis reinforcement [25–27]. Several studies have attempted to investigate these mechanisms. For example, L'Amoreaux used Terminal deoxynucleotidyl transferase dUTP Nick-End Labeling (TUNEL) staining to show that the rate of DNA fragmentation was significantly higher in degenerated salivary glands [28]. In addition, it is reported that caspase3 activity levels increased significantly in tick salivary glands 72 h post-engorgement [29]. Similarly, metabolic activity levels and morphological characters differed substantially among the salivary glands of fast-feeding, engorged ticks, and 3 d post-engorged female brown dog ticks (*Rhipicephalus sanguineus*); metabolic activity levels in the salivary glands of the fast feeding ticks kept in high level, and nuclear and plasma membranes were intact. In engorged ticks, the membranes of the type I and type II acini remained intact, but the nuclei had changed, ATPase activity (proportional to membrane integrity) had decreased, and acid phosphatase activity (inversely proportional to ATPase activity) had increased. At 3 d post-engorgement, salivary gland tissue was severely degraded,

the membranes of most cells (some type I acini, most type II acini, and all type III acini) had disintegrated, ATPase activity had decreased, acid phosphatase activity had increased, and apoptotic bodies had appeared [30]. Thus, during salivary gland degeneration, nuclear changes precede cytoplasmic changes and are accompanied by the formation of apoptotic bodies. Proteomic sequencing indicated that energy-related proteins were downregulated during salivary gland degeneration, but some proteins associated with the degradation of DNA or proteins were up regulated; certain proteins involved in apoptosis or autophagy were also differentially expressed [31]. Thus, similar to some insects, the degradation of tick salivary glands represents programmed cell death mediated by ecdysone [32–36]. There is also a relationship between autophagy and apoptosis during the degeneration of tick salivary glands [37]. In summary, both autophagy and apoptosis are involved in the degeneration of tick salivary glands but their relationship is unclear. Most organisms, from mammals to insects, switch between autophagy and apoptosis [38]. In mammals, autophagy related 5 (ATG5) plays a regulatory role during the transition from autophagy to apoptosis, and the molecular regulation mechanisms involving this protein have been characterized [39]. ATG5 may promote the expansion of autophagosomes by combining ATG12 and ATG16 to form ATG12-ATG5-ATG16L complexes [40]. Alternatively, ATG5 may be cleaved to produce NtATG5 (24 kDa, amino-terminal truncated), which binds to the apoptosis inhibitor protein BCL-XL located on the mitochondrial membrane; this binding leads to the activation of the pro-apoptotic protein BAX to form a BAX-BAX homodimer, thereby promoting apoptosis [39,41]. It was recently shown that, in most insects, the transcription levels of ATG5 and associated genes were the highest during metamorphosis [42,43]. In addition, similar to ATG5 in mammals, insect ATG5 is cleaved by calpains, inducing apoptosis after autophagy [33,42–44]. However, the mechanisms underlying the transition between apoptosis and autophagy in ticks, particularly with respect to salivary gland degeneration, remain unclear.

To address this knowledge gap, we investigated the relationship between autophagy and apoptosis in the salivary glands of the tick *R. haemaphysaloides*. We first reported the concentration-dependent ecdysone-mediated programmed cell death in ticks. RhATG5 was cleaved to produce Nt-RhATG5 during the fast-feeding period. Nt-RhATG5 mediated the transition to apoptosis following 20E-induced autophagy. High 20E concentrations led to high levels of intracellular $Ca^{2+}$, which activated the calpains and mediated the cleavage of RhATG5.

## Materials and methods

### Ethical statement

The care of rabbits and mice was approved by the Institutional Animal Care and Use Committee of the Shanghai Veterinary Research Institute (IACUC approve number: SHVRI-mo-20180306-008; SHVRI-ra-20180415-04; shvri-ZD-2019-027), and authorized by the Animal Ethical Committee of Shanghai Veterinary Research Institute.

### Ticks and animals

Adult *R. haemaphysaloides* were attached to the ears of New Zealand White rabbits (about 3kg) and 30 female *R. haemaphysaloides* and 15 male *R. haemaphysaloides* per ear; rabbits were provided by the Shanghai Laboratory Animals Center (Shanghai Institutes for Biological Science, Chinese Academy of Sciences, Shanghai, China) [45].

## TUNEL staining

The paraformaldehyde-fixed salivary glands were subjected to paraffin-sectioning and antigen retrieval. Sections were then permeabilized with 0.1% Triton X-100 and incubated for 1 h with 1:9 TdT mixed with fluorescent-labeled dUTP at 37˚C, following the instructions of the Roche In Situ Cell Death Detection Kit, POD (Roche, 11684817910). To stain the nuclei, sections were washed three times with phosphate-buffered saline (PBS; 0.14 M NaCl, 0.0027 M KCl, 0.01 M phosphate buffer; pH 7.4)/0.5% Tween-20, and then incubated with 1 μg/mL 4', 6'-dia-midino-2-phenylindole (DAPI, Invitrogen, Carlsbad, CA, USA) in dd $H_2O$ for 20 min. After washing, the sections were mounted using fluorescent mounting medium under glass cover-slips, then viewed and photographed on a Zeiss LSM880 Laser Scanning Confocal Microscope.

## Transmission electron microscopy (TEM) analysis

The glutaraldehyde-fixed sections were ultrathin sectioned and treated using standard TEM procedures [46]. Sections were then viewed and photographed using a Tecnai G2 Spirit BIOT-WIN TEM (FEI, Hillsboro, Oregon, USA).

## Quantitative real-time polymerase chain reactions (qRT-PCRs)

All related genes were identified using a BLAST analysis [47] (http://www.ncbi.nlm.nih.gov/BLAST/). The salivary glands of the *R. haemaphysaloides* allowed to feed for different periods were collected for RNA extraction after microdissection. RNA was converted into the first-strand cDNA using a HiScript III RT SuperMix for qPCR (+gDNA wiper) kit (Vazyme Bio-tech, Nanjing, China), following the manufacturer's protocols. The double-stranded cDNAs of each gene were used as templates for qRT-PCR, along with specific primers (S1 Table), which were designed using Primer Premier 5. qRT-PCRs were performed using ChamQ Universal SYBR qPCR Master Mix (Vazyme Biotech) green and gene-specific primers with a QuantStu-dio 5 System PCR System (Applied Biosystems, Austin, TX, USA). The qRT-PCR cycling parameters were 95˚C for 30 s, followed by 40 cycles of 95˚C for 5 s and 60˚C for 30 s. All sam-ples were analyzed three times.

The data were normalized to the gene expression of elongation factor-1 (ELF1A, Genbank accession no. AB836665) [48] using the $2^{-\Delta\Delta Ct}$ method [49,50]; ΔCt was calculated by subtract-ing the average ELF1A Ct value from the average Ct value of the target gene.

## Sequencing and cloning of RhATG5

RhATG5-specific primers (S2 Table) were designed based on a comparison of the salivary gland transcriptomes of fasted and engorged *R. haemaphysaloides* [51]. A BLAST analysis of the translation products deduced from the open reading frames (ORFs) was performed. We used ExPASy (https://web.expasy.org/compute_pi/) to predict the molecular weight and iso-electric point (PI) of RhATG5. We aligned RhATG5 with the ATG5 protein sequences of vari-ous organisms using Genetyx ver. 6 (GENETYX, Tokyo, Japan). For phylogenetic analysis, the alignment of the sequences was performed using the MUSCLE algorithm [52] and inferred using the Maximum likelihood method with the default settings in MEGA X software [53]. Bootstrap support values were estimated using 500 bootstrap replicates. The sequences were obtained from GenBank or UniProt.

## RhATG5 cleavage assay

The ORF sequence of RhATG5 was subcloned into the pET-30a prokaryotic expression vector, using In-Fusion HD Cloning Kits (Takara Clontech, Mountain View, CA, USA). We

constructed the mutant and the mutation primers for RhATG5 using the Quick Mutation Gene Site-Directed Mutagenesis Kit (Beyotime, Shanghai, China), following the manufacturer's instructions. The specific mutation primers that we designed are listed (S2 Table). The recombinant plasmids pET-30a-RhATG5 and pET-30a-RhATG5$^{191-199\Delta}$ were constructed and transformed into BL21 (DE3) competent *Escherichia coli* cells (Tiangen, Beijing, China). To induce the expression of the recombinant proteins RhATG5 and RhATG5$^{191-199\Delta}$, BL21 cells were incubated in a final concentration of 1 mM isopropyl thio-β-D-galactoside (IPTG) at 25°C for 12 h. Then, recombinant RhATG5 and RhATG5$^{191-199\Delta}$ were affinity-purified using Ni-NTA His•Bind Resin (Merck-Millipore, Darmstadt, Germany). Recombinant RhATG5 and RhATG5$^{191-199\Delta}$ were then incubated with 10μg μ-calpain (Merck) in TE buffer (10 mM Tris-HCI, 1 mM EDTA, PH 8.0) at 25°C for 3 hours. The reaction was stopped by adding 2 × SDS loading buffer. The mixtures were heated at 100°C for 10 min and then separated on GenScript ExpressPlus PAGE Gels 10 × 8, 12%, 10 wells (Genscript, Nanjing, China). Gels were stained with Coomassie brilliant blue using eStain L1 (Genscript).

## *In vitro* culture of tick salivary glands and 20E treatment

Unfed ticks were sterilized in 70% ethanol for 30 s and then dried with filter paper. The salivary glands of each tick were removed using microdissection, immediately transferred to *L15* medium supplemented with 1% penicillin-streptomycin at 27°C, and cultured *in vitro* as previously described [54]. Cultured salivary glands were then treated with 1 μM, 5 μM, 10 μM, or 20 μM 20E (Sigma-Aldrich, St. Louis, MO, USA). Incubation was stopped and samples were collected at 24 or 48 h. Dimethyl sulfoxide (DMSO; Sigma-Aldrich) was used as the control. Each treatment group included the salivary glands from 10 ticks. After incubation, a pair of the salivary glands in each group was fixed in 4% paraformaldehyde for TUNEL staining or fixed in 2.5% glutaraldehyde for at least 24 hours at 4°C for TEM. The remaining salivary glands were used for western blots.

## Antibody generation, western blotting and indirect immunofluorescence antibody (IFA)

Recombinant his-tag RhATG5 was expressed and purified according to standard protocols [55]. Polyclonal antibodies (pcAb) against recombinant RhATG5 were developed in mice. We tested the specific recognition of the recombinant protein by the antibodies using western blots. To distinguish between autophagy and apoptosis, we chose the molecular markers RhATG8 (autophagy, MN395580) and for antibody production and the Rhcaspase7 pcAb was prepared in our lab [56] (apoptosis, MN395579). We predicted the epitopes of RhATG8 online (http://www.iedb.org/), and synthesized Keyhole limpet hemocyanin (KLH) [57]-coupled polypeptides based on the epitope amino acid sequences (MKFQYKEEHPFEK). Polypeptides were dissolved in PBS and Freund's adjuvant (complete and incomplete; Sigma-Aldrich). Equal volumes of these solutions were then emulsified together and injected into 6–8 week-old BALC/c mice or New Zealand White rabbits provided by the Shanghai Laboratory Animals Center (Shanghai Institutes for Biological Science, Chinese Academy of Sciences, Shanghai, China). Injections were repeated three times at two-week intervals. Serum samples were collected seven days after the third injection.

For western blots assays, total proteins from the salivary glands of 10 ticks were extracted using Tris-buffered saline (TBS; 10 mM Tris–HCl, pH 7.5; 150 mM NaCl with 1 mM phenyl-methanesulfonyl fluoride) to eliminate individual differences. Protein concentrations were quantified using the Bradford Protein Assay Kit (Beyotime), following the manufacturer's instructions. We used 20 μg of protein from each sample for SDS-PAGE. Western blotting and

IFA were then performed as previously described [55], and images were captured by ChemiDoc Touch (Bio-rad, Hercules, CA, USA) or Odyssey (*LI-COR*, Nebraska, USA) Imaging System. The intensity of target protein bands was measured by Image-Pro Plus software (Media Cybernetics). The anti-alpha tublin primary antibody was purchased from Proteintech (Rosemont, IL, USA). The Goat anti-Mouse IgG (H+L) Secondary Antibody conjugated with HRP, Goat anti-Rabbit IgG (H+L) Secondary Antibody conjugated with HRP, Goat anti-Rabbit IgG (H+L) Highly Cross-Adsorbed Secondary Antibody, Alexa Fluor Plus 594, Goat anti-Mouse IgG (H+L) Highly Cross-Adsorbed Secondary Antibody, Alexa Fluor Plus 488 were all purchased from Invitrogen.

## Measurement of cellular $Ca^{2+}$ in salivary glands

All salivary glands were washed three times with PBS, and then incubated with 3 μM AM ester Calcium Crimson dye (Invitrogen) for 30 min at 27°C. After incubation, the salivary glands were washed three times with PBS, then incubated with 1 mg/mL 2,5'-Bi-1H-benzimidazole, 2'-(4-ethoxyphenyl)-5-(4-methyl-1-piperazinyl)- 23491-52-3 (Hoechst 33342, trihydrochloride, trihydrate, Life Technologies, H3570) in water for 20 min. After washing three times with PBS, each whole salivary gland was placed on a glass slide, sealed with Lab Vision PermaFluor (Thermo Scientific, Carlsbad, CA, USA), and observed under a Zeiss LSM880 Laser Scanning Confocal Microscope (Oberkochen, Gemany).

## RNAi *in vivo* and *vitro*

Specific interference primers were designed based on the known sequences (S2 Table), adding the T7 polymerase promoter sequence to the 5' end of each primer. After PCR amplification, the amplicons were purified to obtain templates for double-strand RNA (dsRNA) synthesis. The dsRNAs of RhATG5 and Luciferase were generated using the T7 RiboMAX Express RNAi system (Promega, Madison, WI, USA). The dsRNA of each gene was either microinjected into unfed female ticks (1 μg/ tick).

Our preliminary results indicated that the dissected salivary glands of female *R. haemaphysaloides* were viable in culture (see above). This indicated that salivary gland cultures were suitable for a detailed study of RhATG5 RNAi *in vitro*. We used salivary glands extracted from ticks during the fast-feeding period (5 and 7 days) for *in vitro* RNAi experiments. Salivary gland cultures were treated with RhATG5 dsRNA (1 μg/mL) for 48 h. After 48 h, ticks and salivary glands were analyzed.

## Data analysis

GraphPad PRISM 6.0 software (La Jolla, CA, USA) was used for all data analyses. Mean ± standard error (SEM) values were calculated for three independent experiments, and two-tailed Student's *t* tests were used to identify significant differences between groups (*p < 0.05; **p < 0.01, ***p < 0.001, ****p < 0.0001).

# Results

## Both autophagy and apoptosis occurred in the degeneration of salivary glands

To determine the exact time of tick salivary gland degeneration, we observed the morphologies of the salivary glands of ticks fed for different periods. The length, width and bubble diameter of the salivary glands increased continuously from the first feeding day until engorgement and rapidly degenerated and atrophied after engorgement (Fig 1A and 1B). TEM observations

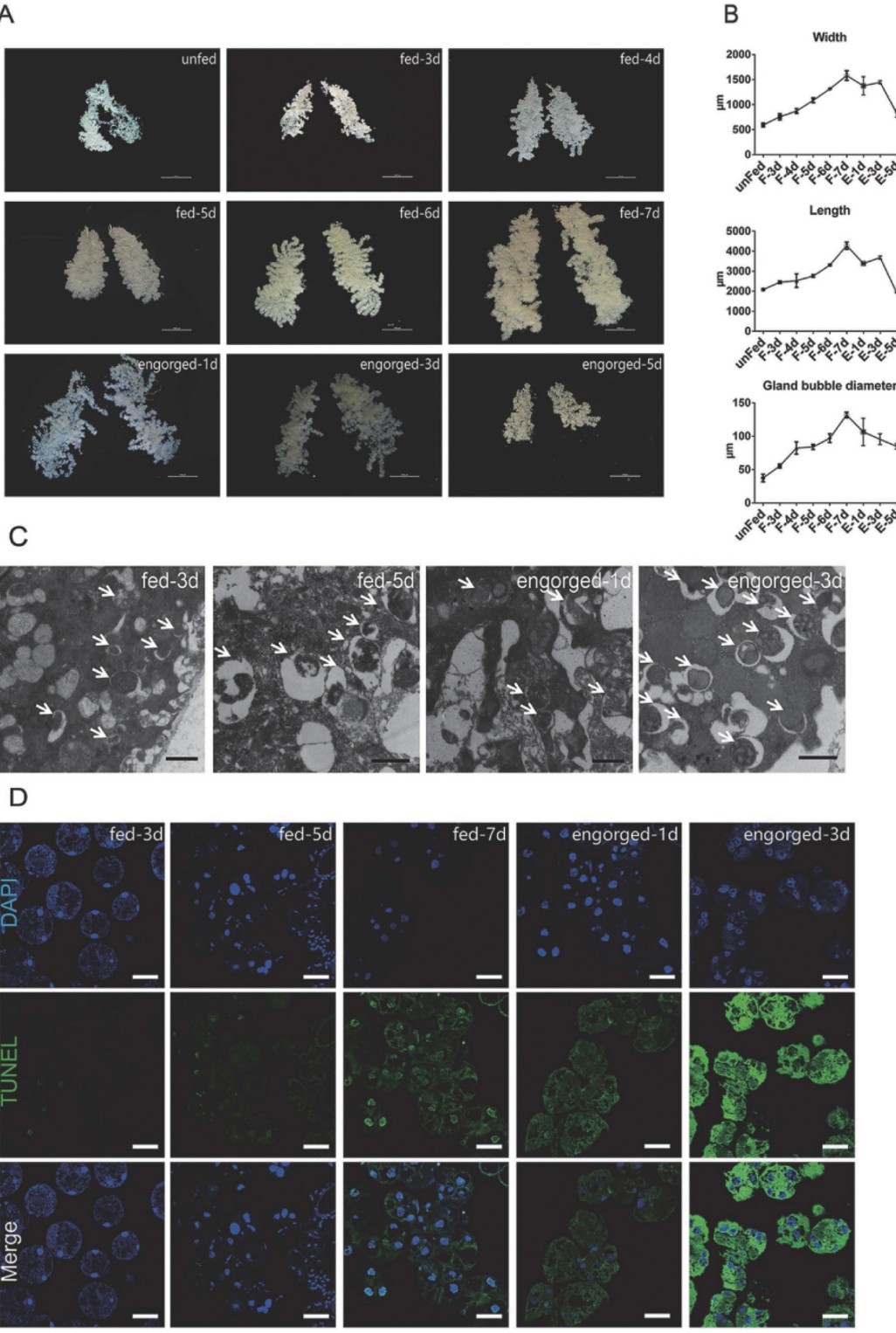

**Fig 1. Autophagy and apoptosis profile of tick salivary glands at different feeding times.** (A) Morphological observation *in vitro* (scale bar: 1000 μm) of salivary glands at different feeding times. (B) Analysis of *A*), Bars represent mean ± SD for three replicates. (C) TEM observation (scale bar: 1 μm) of salivary glands at different feeding times. Autophagosomes are marked with white arrows. (D) TUNEL staining (green, scale bar: 50 μm) of salivary glands during different feeding periods. Please see S1 and S2 Data for the numerical values and original images used in Fig 1.

showed that mitochondria large amounts of bilayer membrane autophagosomes and autophagic vacuoles during the late feeding and post engorged periods (Fig 1C). Compared to the early feeding period, DNA fragmentation increased from the fast feeding period to post-engorgement, along with the emergence of autophagosomes (Fig 1D). The TUNEL positivity of cytosol may relate to the presence of a large number of mitochondria. Autophagy and apoptosis occur during the degeneration of tick salivary glands, which is consistent with previous observations [4,16,30,58,59].

## Relative expression levels of autophagy and apoptosis genes in tick salivary glands of ticks fed for different periods

The cDNA of tick salivary glands at different feeding times were subjected to qRT-PCR to evaluate the expression profiles of autophagy and apoptosis related genes. To investigate the expression patterns of ATG homologs associated with autophagy, a total of 13 putative RhATGs, including RhATG3, RhATG4B, RhATG4D, RhATG5, RhATG6 (BECN1), RhATG7, RhATG8, RhATG9, RhATG10, RhATG12, RhATG13, RhATG14 and RhATG16, occur in the salivary gland transcriptome of *R. haemaphysaloides* [51,56], all of the RhATGs contained classic ATG domains. Most autophagy-related genes were upregulated between the 1st and 3rd day of feeding, and then down-regulated during the fast-feeding period (days 4–7; Fig 2A). Post-

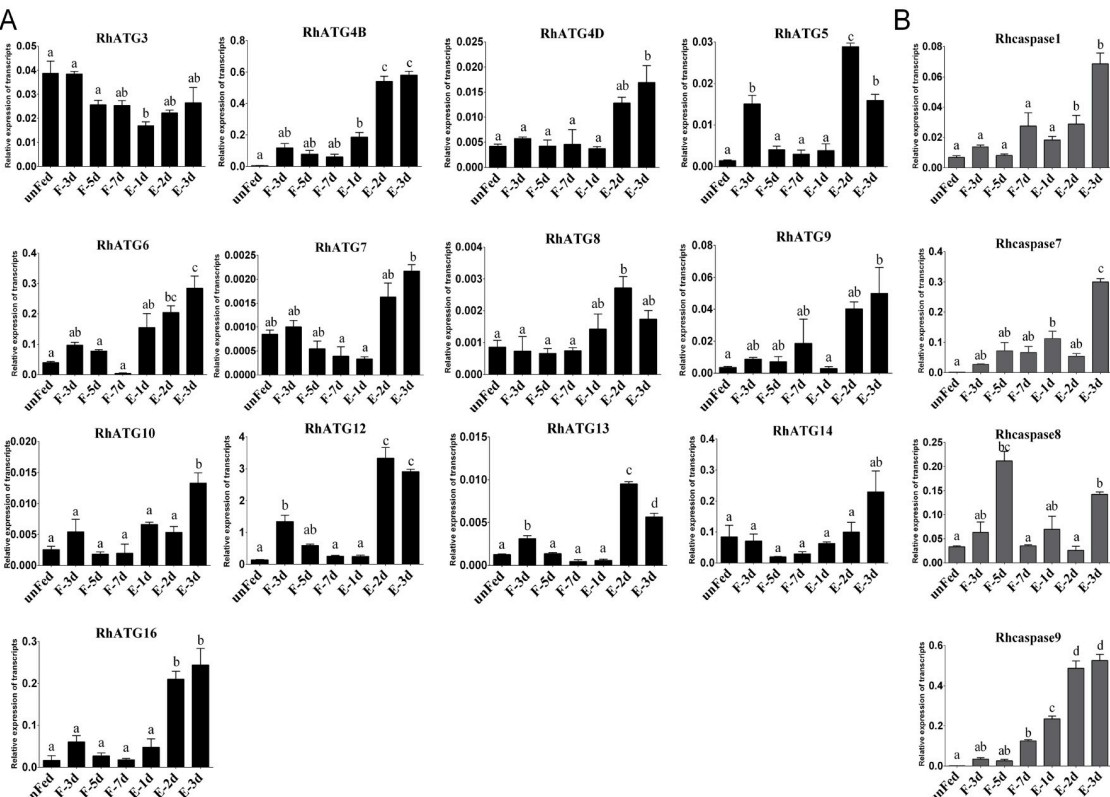

**Fig 2. Transcription profiles of autophagy and apoptosis related genes.** (A) autophagy and (B) apoptosis, as quantified using qRT-PCR. Expression levels of all genes were measured in the salivary glands of female ticks at various stages during the feeding process: unfed to engorged to 1–3 d post engorgement, F: fed, E: engorged. Each group included 10 ticks and each experiment was repeated three times. Please see S1 Data for the numerical values used in Fig 2. Bars represent mean ± SD for three replicates. Significance of differences determined by one-way ANOVA. Data not sharing a common letter indicated there was a significant difference ($p<0.05$).

engorgement, most of autophagy-related genes was sharply upregulated except RhATG3, but the reason remains to be explored (Fig 2A).

 Using the same strategies, we identified 4 predicted Rhcaspases, including Rhcaspase1, Rhcaspase7, Rhcaspase8, and Rhcaspase9. All 4 Rhcaspases contained the conserved peptide sequence of QACR(I)G [37]. All apoptosis-associated genes were upregulated in fast feeding period and post-engorged period compared to slow feeding period (Fig 2B). RhCasapase8 was also sharply upregulated on day 5 of feeding (Fig 2B).

### Cloning and recombinant expression of RhATG5

Based the results above, RhATG5 was cloned to study the interaction between autophagy and apoptosis. The *R. haemaphysaloides* ATG5 homolog was designated RhATG5 (GenBank accession no. MK841509). This is the first reported ATG5 homolog in ticks. The full-length open reading frame (ORF) of RhATG5 was 810 base pairs long and encoded a 270 amino acid peptide. The predicted molecular weight of the RhATG5 protein was 31.3 kDa, with a theoretical pI of 6.19. Homology analyses indicated that RhATG5 is a member of the ATG5 family. The APG5 domain extends from amino acids 81 to 270 (marked with a black line), and has 54.58%, 53.85%, 59.85%, and 50.19% identity with the ATG5 proteins from (*Homo sapiens*; NP_004840.1), mice (*Mus musculus*; NP_444299.1), fruit flies (*Drosphilia melanogaster*; NM_132162.4), and the cotton bollworm (*Helicoverpa armigera*; AMQ76404.1), respectively (S1A Fig). The lysine (K) 130 of Homo-ATG5 is necessary for covalent bond formation in the ATG12-ATG5 complex [60]. This critical lysine residue of ATG5 is present at position 132 in RhATG5 (S1A Fig, marked with black frame). Amino acids His80 and Ser127 of homo-ATG5 (His82 and Ser129 in RhATG5) (S1A Fig, marked with black frame) play important roles in the formation of the ATG12-ATG5-ATG16 complex [60]. Although ticks are evolutionarily distant from other animals, our phylogeny of ATG5 homologs from a variety of species suggests a close relationship among ATG5 homologs (S1B Fig). This indicates that autophagy-associated genes, such as the ATG5 homologs, are ancient and have been conserved over long periods of evolutionary time.

### RhATG5 was cleaved and intracellular calcium increased in tick salivary glands during the fast feeding phase

RhATG5, RhATG8 and Rhcaspase7 were tested to show the occurrence of autophagy and apoptosis. RhATG5, with a molecular weight of 33kDa, kept increasing during all the feeding times until engorged. RhATG5 was cleaved into NtATG5 (represented a fragmented form of RhATG5 with a molecular weight of 25 kDa) from the fast-feeding to the engorgement phases (Fig 3A and 3B). RhATG5 and NtATG5 decreased rapidly post engorgement. The expression level of RhATG8-PE (16 kDa) appeared at the early feeding time point and low expressed during the fast feeding periods, and then peaked post-engorgement, consistent with our qRT-PCR results (Fig 3A and 3B). Pro-Rhcaspase7 (34 kDa) increased during fast feeding periods and kept the increase post engorged. Cleaved-Rhcaspase7 (17 kDa) emerged followed the decrease of RhATG8-PE and kept increasing until post engorgement (Fig 3A and 3B).

 The $Ca^{2+}$ concentrations of tick salivary glands during different feeding periods were studied to display the concentration changes of $Ca^{2+}$ in the degeneration of tick salivary glands. According to the results, $Ca^{2+}$ concentration (displayed as red fluorescence) in the tick salivary glands increased between the fast-feeding and the engorgement phases along with the degeneration of tick salivary glands (Fig 3C); the transcription levels of two Rhcalpains also increased during this period (Fig 3D), which is consistent with the appearance time of Nt-RhATG5. Our

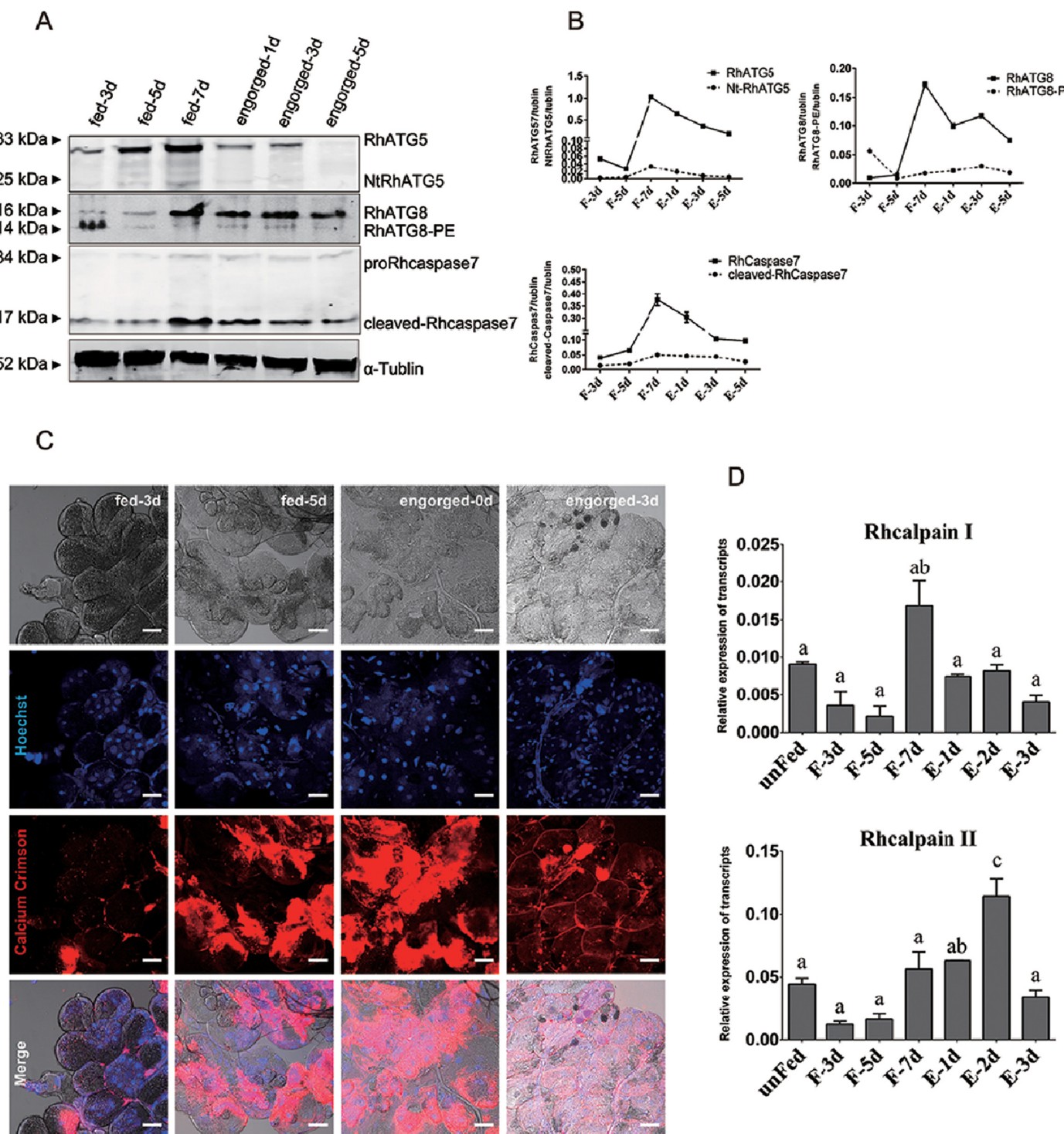

**Fig 3. RhATG5 cleaved and intracellular calcium increased during the fast feeding phase.** (A) Relative protein expression of RhATG5 and other proteins associated with autophagy and apoptosis related genes. The salivary gland samples were harvested at feeding days 3, 5, 7 and post engorgement days 1, 3, and 5, following by immunoblotting with the anti-RhATG5 (pcAb), anti-RhATG8 (pcAb), anti-Rhcaspase7 (pcAb) and anti-α-tublin (mAb). (B) Statistical analysis of (A). (C) Ca$^{2+}$ concentration of tick salivary glands in different feeding periods was measured using Calcium Crimson dye (Invitrogen), and is represented by red fluorescence, scale bars: 50 μm. (D) qRT-PCR quantification of Rhcalpains expression in the salivary glands of female ticks at various stages during the feeding process: unfed to engorged to 1–3 d post engorgement. All experiments were performed in triplicate; bars represent the mean ± S. D, and Significance of differences determined by one-way ANOVA. Data not sharing a common letter indicated there was a significant difference ($p < 0.05$). Please see S1 and S2 Data for the numerical values and original images used in Fig 3.

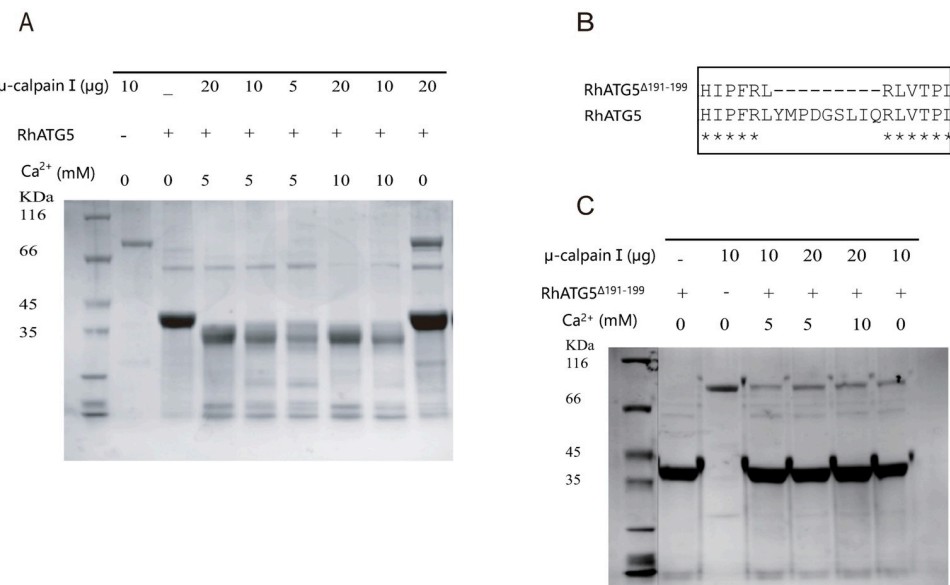

**Fig 4. *In vitro* RhATG5 cleavage assays.** (A) SDS-PAGE showing that recombinant RhATG5-His is cleaved by μ-calpain and is not cleaved by calpain in the absence of $Ca^{2+}$. (B) The RhATG5 cleavage site. (C) The mutant recombinant RhATG5$^{191-199Δ}$-His is not cleaved by μ-calpain. Please see S2 Data for the original images used in Fig 4.

results indicated that the cleavage of RhATG5 may be caused by an increase in intracellular calcium concentration.

## RhATG5 was cleaved by μ-calpains *in vitro*, depending on calcium concentration

The position of the 6-amino-acids (QTTTER) is crucial for the calpain cleavage site (marked with red frame) of Homo-ATG5 [39], which occurs at positions 192 to 197 in RhATG5. One calpain cleavage site was predicted in our analyses of RhATG5 homology (Fig 4A). The coding sequences of RhATG5 and RhATG5$^{191-199}$were cloned into prokaryotic expression vectors (pET-30a) to produce recombinant Rhcaspases 7, 8, and 9. All of the recombinant proteins were expressed in *E. coli*. After purification, His-RhATG5 (39 kDa) was incubated with μ-calpain at different concentrations, $Ca^{2+}$ treated RhATG5 was cleaved into one active fragment (NtRhATG5, 28 kDa) (Fig 4A). With the same $Ca^{2+}$ concentration and the same reaction time, along with the increased concentration of μ-calpain, the bands of NtRhATG5 were more obvious, and under the same concentrations of μ-calpain, changing the $Ca^{2+}$ concentration did not affect the cleavage of RhATG5 (Fig 4A). After the cleavage site was deleted (Fig 4B), the recombinant protein RhATG5$^{191-199Δ}$ was not cleaved by μ-calpain under identical experimental conditions (Fig 4C).

## Induction of autophagy and apoptosis in tick salivary glands by culturing with 20E

The similarity of results between the 20E titer and the expression profile of autophagy and apoptosis genes in tick salivary glands at different feeding times indicated that 20E might induce degeneration of the salivary glands. Salivary glands of unfed ticks were cultured with different concentrations of 20E *in vitro*. The unfed tick salivary glands exhibited positive TUNEL

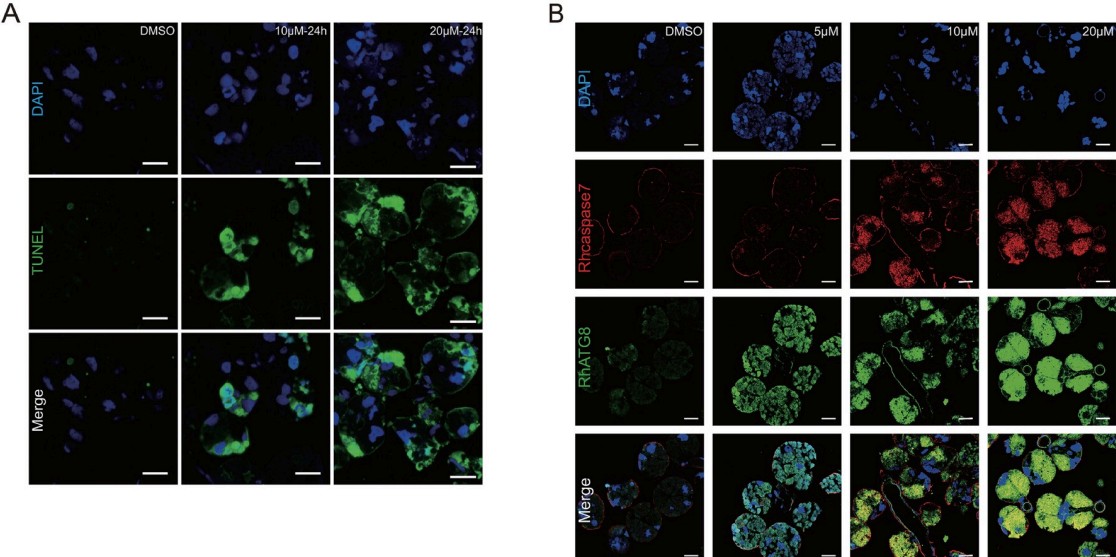

**Fig 5. Programed cell death caused by 20E *in vitro*.** (A) TUNEL staining showing green fluorescence, Scale bars: 50 μm. (B) Rhcaspase7 (red fluorescence; pcAb from rabbits) and RhATG8 (green fluorescence; pcAb from mice) were co-localized after treatment with high concentrations of 20E (10–20 μM; displayed as yellow fluorescence), scale bar: 20 μm. Please see S2 Data for the original images used in Fig 5.

staining after treatment with 10 μM 20E at 24 h and 48 h and apoptosis increased with time (Fig 5A). After incubated with 5μM 20E, ATG8 (green fluorescence) increased at 24h, along with the concentration of 20E, immunofluorescence analysis showed that RhATG8 and Rhcaspase7 (red fluorescence) were co-localized after treatment with high concentrations (10 μM and 20 μM) of 20E (Fig 5B; displayed as yellow fluorescence).

The qRT-PCRs demonstrated that the expression profiles of ATGs were upregulated (p<0.0001) 24 h after treatment with 10 μM 20E. These results were consistent with western blot results. All ATGs increased during the first 48 h of feeding but significantly decreased at 72 h (Fig 6A). The apoptosis-related genes, Rhcaspase7 and Rhcaspase8 were upregulated at 24 h, similar to the ATGs (at 24 h; Fig 6A and 6B); however, the expression levels of these genes decreased between 24 and 48 h. All the Rhcaspases were upregulated at 72 h (Fig 6B).

## Treatment with high concentrations of 20E increased intracellular Ca²⁺ and mediated the cleavage of RhATG5 in tick salivary glands

As mentioned above, unfed tick salivary glands were treated with different concentrations of 20E. The results of western blots showed that RhATG8 and RhATG8-PE was upregulated after treatment with a low concentration (5 μM) of 20E. With the increase in the concentration of 20E, RhATG8 and RhATG8-PE decreased and both cleaved-Rhcaspase7 and NtRhATG5 increased after treatment with high concentrations of 20E (10 μM and 20 μM; Fig 7A and 7B).

Ca²⁺ was tested in the 20E treated tick salivary glands to demonstrate the increased Ca²⁺ during the ecdysone mediated degeneration of tick salivary glands (Fig 7C). Similarly, RhcalpainI peaked at 24 h, while RhcalpainII expression peaked at 72 h (Fig 7D). The data shows that the cleavage of ATG5 and caspase-3 depends on the increase of the calcium concentration after 20E induction.

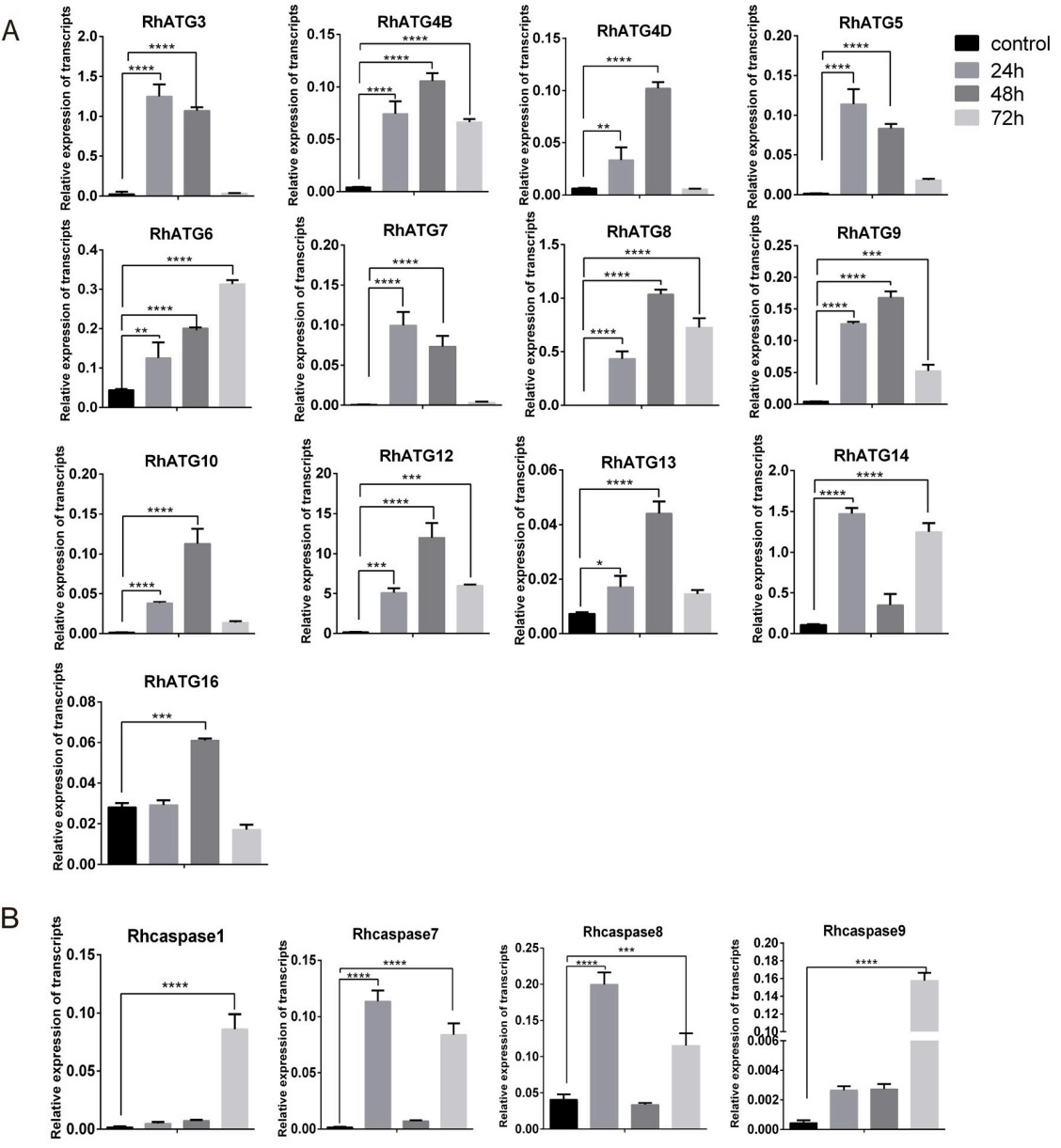

**Fig 6. qRT-PCR quantification of the expression levels of (A) autophagy-related genes and (B) apoptosis-related genes after treatment with 20E.** Please see S1 Data for the numerical values used in Fig 6. Bars represent mean ± SD for three replicates. Significance of differences determined by Student's *t*-test: *p < 0.05; **p < 0.01, ***p < 0.001, ****p < 0.0001.

### Inhibition of RhATG5 delayed apoptosis in the tick salivary glands

RhATG5 was knocked down in tick salivary glands *in vitro* (Fig 8A, S2A and S2B Fig), and the results showed that RhATG8-PE was downregulated relative to the control at 1 d post-engorgement and cleaved-Rhcaspase7 was downregulated relative to the control at day 5 of fast feeding (Fig 8A). RhATG5 knockdown result in the inhibition of apoptosis, and the degree of nuclear and cytoplasmic separation was reduced (Fig 8B). In addition, TEM analysis showed that, compared with the control groups, the degree of vacuolization was noticeably reduced

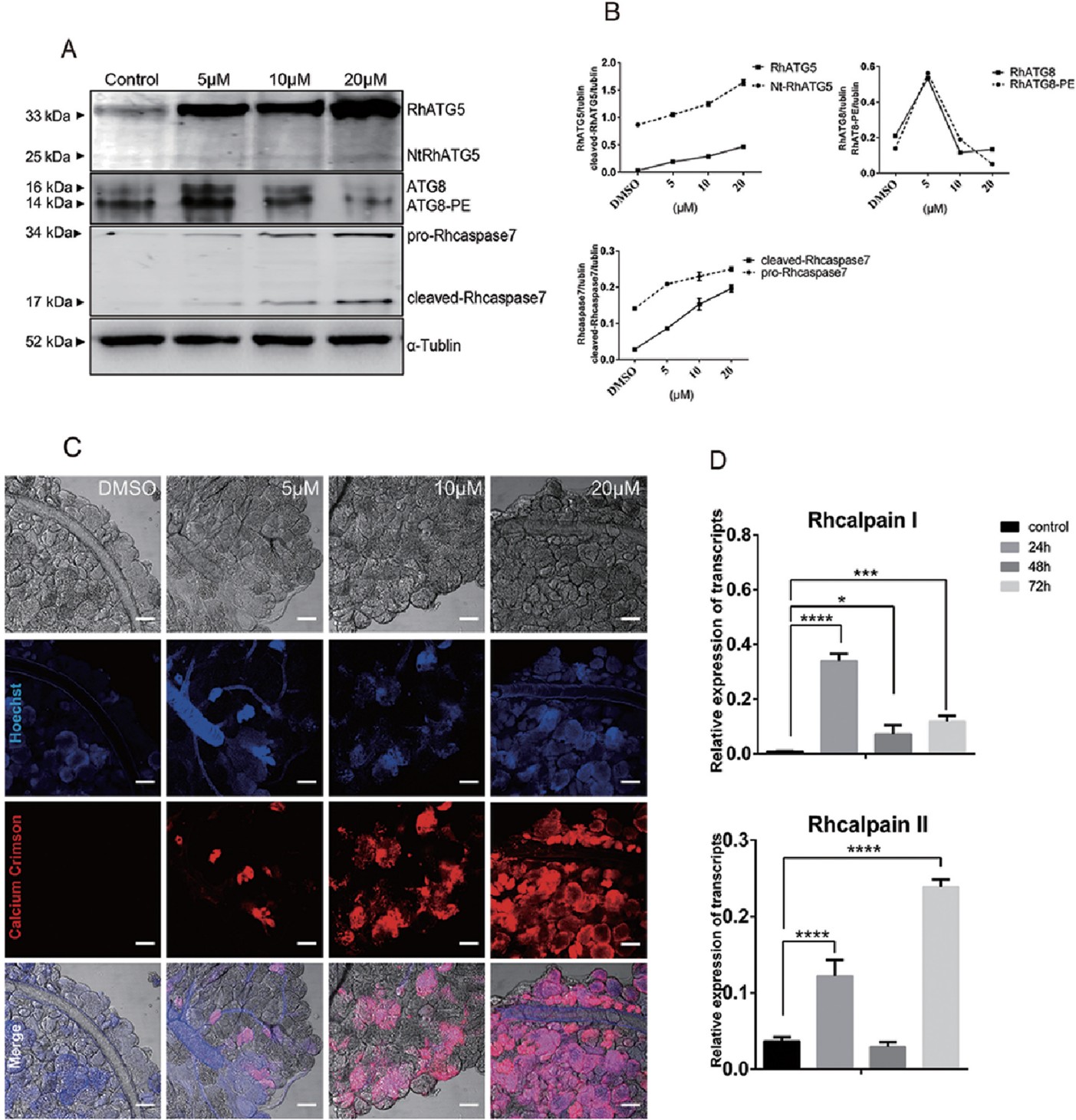

**Fig 7. Treatment with 20E increased cellular Ca²⁺ concentration and mediated the cleavage of RhATG5 *in vitro*.** (A) Western blots showing the expression levels of proteins associated with autophagy and apoptosis after treatment with different concentrations of 20E, each group consist of the salivary glands of 10 ticks. (B) Statistical analysis of (A). (C) Cellular concentrations of Ca²⁺, using Calcium Crimson dye (red fluorescence, scale bars: 50 μm). (D) qRT-PCR quantification of calpains after treatment with 20E. Bars represent mean ± SD for three replicates. Significance of differences determined by Student's *t*-test: *p < 0.05; **p < 0.01, ***p < 0.001, ****p < 0.0001. Please see S1 and S2 Data for the numerical values and original images used in Fig 7.

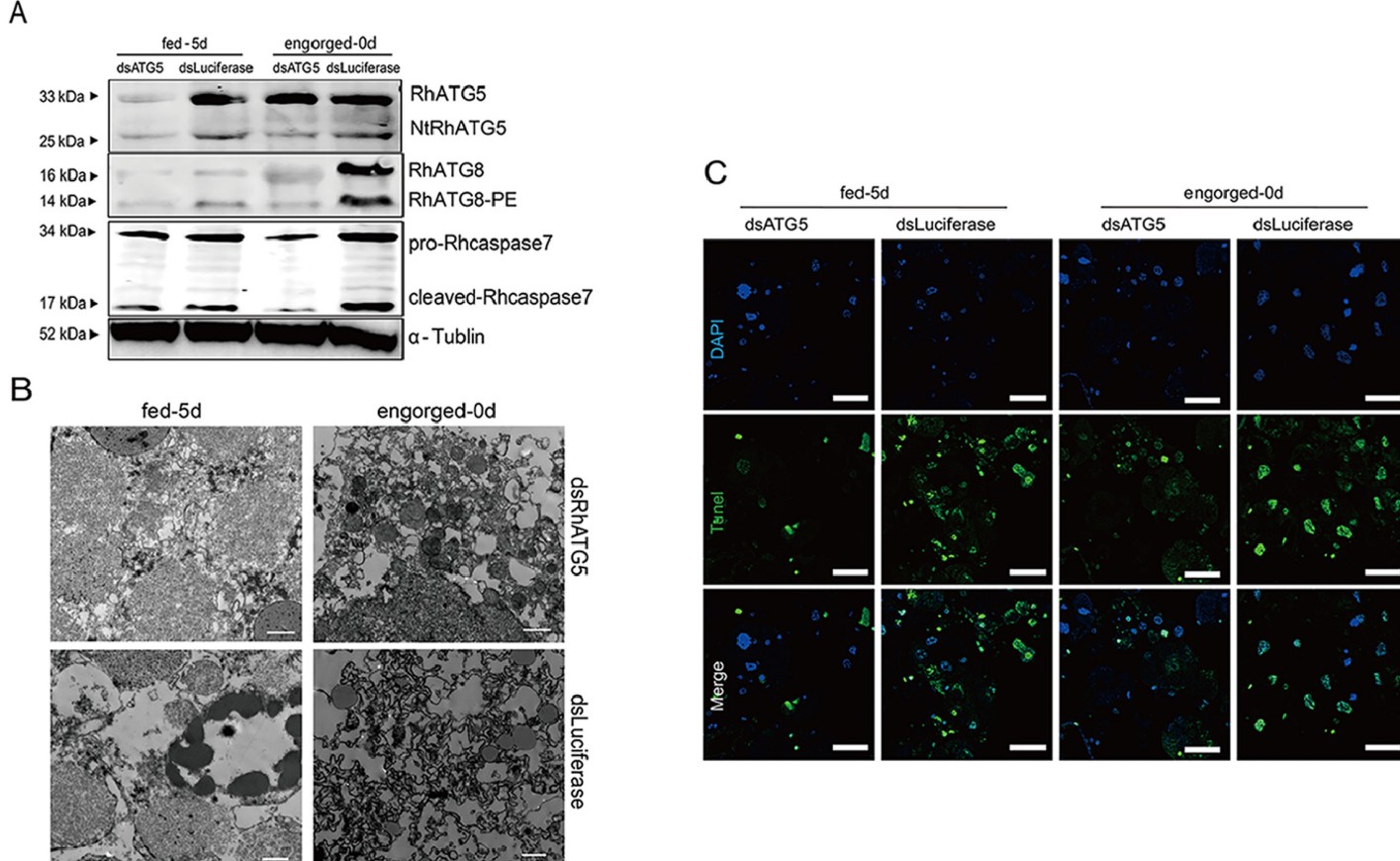

**Fig 8. RNAi of RhATG5 *in vitro*.** (A) Knockdown of RhATG5 inhibited the protein expression of RhATG8-PE and cleaved-Rhcaspase7. (B) Knockdown of RhATG5 reduced the degree of vacuolization in the tick salivary glands, scale bars: 1 μm. (C) Positive TUNEL staining decreased after RNAi of RhATG5, scale bars: 50 μm. Please S2 Data for the original images used in Fig 8.

after knockdown (Fig 8C). These results revealed that RhATG5 is crucial for both autophagy and apoptosis.

### RNAi knockdown of RhATG5 in vivo inhibited tick blood-feeding behavior

RNAi of RhATG5 *in vivo* caused inhibition on the blood-feeding behavior of *R. haemaphysaloides*. When RhATG5 was knocked down (Fig 9A), *R. haemaphysaloides* blood-feeding behavior was inhibited (Fig 9B). In addition, RhATG5-knockdown ticks had a decreased engorgement rate and a higher mortality rate than wild-type ticks (Table 1). However, there was no difference in attachment rate at 48 h between the wild-type and the RhATG5-knockdown groups (Table 1). These results suggest that RhATG5 plays an important role during tick feeding.

### Discussion

Ticks are well-known pathogen vectors which transmitted virus, bacterial and protozoan and they are evolutionarily distant from mammals and insects. Understanding the tick

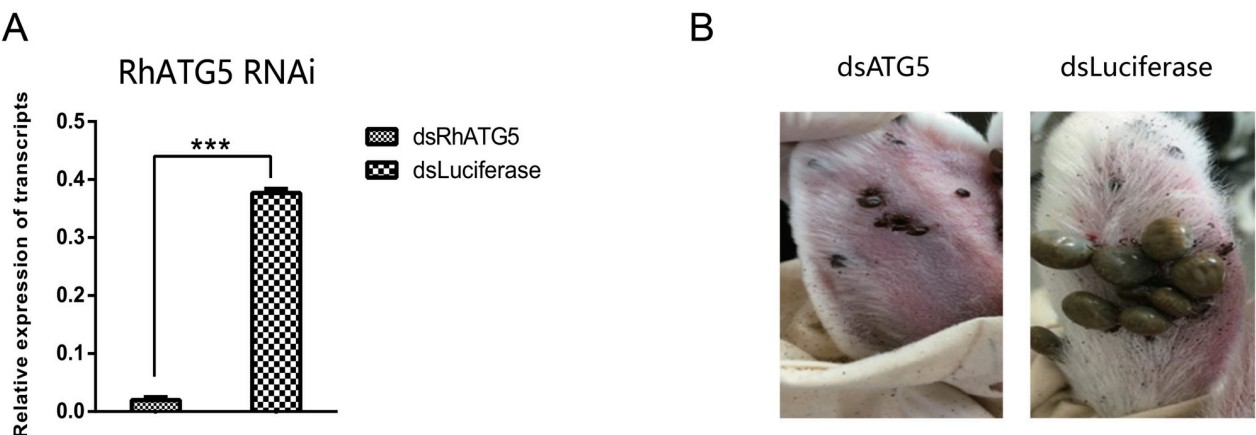

**Fig 9. RNAi of RhATG5 *in vivo*.** (A) Confirmation of RhATG5 silencing using qRT-PCRs. Total RNA was extracted on feeding day 5 from female ticks injected with dsRNA. Bars represent mean relative expression of RhATG5 across three replicates; error bars represent standard error. ***, $P < 0.001$, based on two-tailed Student's *t* tests. (B) Representative images comparing ticks injected with dsRhATG5 with control groups on day 6 of feeding. DsRhATG5 injection inhibited blood-feeding behavior; injected ticks were smaller than control ticks. Please see S1 Data for the numerical values used in Fig 9.

physiological characteristics is the key step for investigating the mechanism of pathogen transmission in ticks. However, there are knowledge gap remained in tick physiology. ATG5 is required for autophagy and apoptosis in most species. However, this was previously not recognized in ticks. The 20E steroid hormone promotes programmed cell death in insects but the mechanism in ticks is unclear. We first proved that 20E promotes both autophagy and apoptosis and that ATG5 is necessary for autophagy and apoptosis in ticks.

Most of the apoptosis and autophagy associated genes were sharply upregulated post engorgement. This indicated that both autophagy and apoptosis occur during tick salivary gland degeneration. ATG5 and ATG12, which are generally thought to form a covalently-bound complex during autophagy [61,62], had similar transcriptional profiles in *R. haemaphy-saloides* during feeding (Fig 2A). ATG16, other major members of the ATG5-ATG12-ATG16 complex [63,64], ATG10, the E2 ubiquitination protein, as the catalyst of complex formation [61,65], were also upregulated after engorgement. Interestingly, the expression levels of most autophagy-related genes in the tick salivary glands peaked during the slow-feeding phase (Fig 2A), as did the ecdysone titer changed. Thus, the upregulation of autophagy-related genes during the slow-feeding period (days 1–4) might be associated with the elevated concentrations of ecdysone in salivary glands, but this supposition requires further experimental confirmation. Our results clearly showed that, during the fast-feeding phase (days 5–7), autophagy-related genes were downregulated, but apoptosis-related genes were upregulated. In addition, the two initial Rhcaspases (Rhcaspase8 and Rhcaspase9) were upregulated before the two effector Rhcaspases (Rhcaspase1 and Rhcaspase7) molecules (Fig 2B). Thus, our results indicated that both autophagy and apoptosis are involved in the degeneration of tick salivary glands.

**Table 1. Effects of knocking down RhATG5 on tick feeding behavior.**

|  | dsRhATG5 | dsLuciferase |
|---|---|---|
| Attachment rate at 48h (%) | 80.22±0.906 | 87.70±0.690 |
| Engorgement rate (%) | 0 | 75.79±3.255 |
| Engorged tick weight (mg) | —— | 363.01±30.01 |

It has been suggested that the peak in hemolymph ecdysteroid concentration during the fast-feeding phase contributes to the initiation of salivary gland degeneration [19]. This hypothesis was supported by the results of the TUNEL and TEM assays of the salivary glands from ticks allowed to feed for different periods (Fig 1A, 1B and 1C). As shown above, the protein NtATG5, which was a maker for the transition from autophagy to apoptosis, was initially expressed during the fast-feeding phase (days 5–7) and was significantly downregulated post engorgement. After a transient peak in expression on day 3 of feeding, the transcriptional levels of most autophagy-related genes were reduced. In contrast, the expression levels of the Rhcaspases increased during the fast-feeding period (Fig 2A and 2B). In addition, intracellular $Ca^{2+}$ concentration and Rhcalpain expression peaked simultaneously with the appearance of NtATG5 (Fig 4C and 4D). The knockdown of RhATG5 *in vitro* delayed both autophagy and apoptosis, demonstrating that RhATG5 is closely associated with both autophagy and apoptosis in the tick salivary glands.

In insects, programmed cell death is usually induced by ecdysone [33,34,66]. In addition, peak ecdysone titer peaks simultaneously with the autophagy-related gene transcription in most insects [32,34,66]. After insects enter metamorphosis, ecdysone concentration in the hemolymph increases sharply and up to 0.15μM, and is converted into highly-active 20E; 20E then induces the degeneration of obligate tissues [34,67]. It has been reported that the ecdysone titer (about 0.2μM) in tick salivary glands increases throughout the feeding period, and remains elevated for up to six days post engorgement [19]. However, it is unclear whether these events are associated in ticks.

For the 20E treatment *in vitro*, considering the differences in activity between commercial 20E and 20E synthesized in the body, and there may be differences in tick species and measurement errors in the preliminary concentration detection, we mainly refer to the concentration of 20E in insects. Our results showed that, similar to insects [43,68], 20E induced

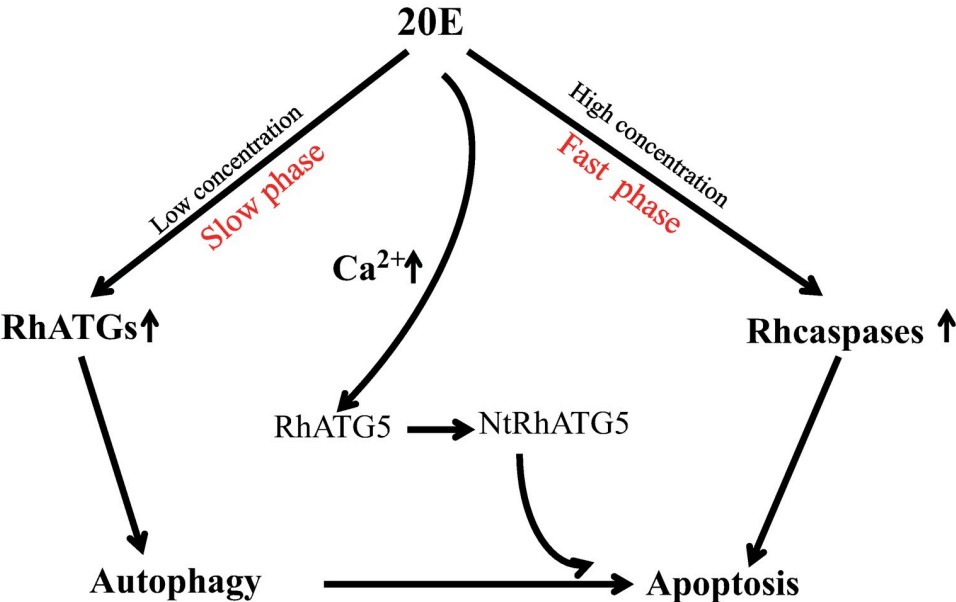

**Fig 10. Schematic showing how 20E induces autophagy and apoptosis in the tick salivary glands.** At low concentrations (during the slow-feeding phase, days 1–4 of feeding) 20E induces autophagy. As the concentration of 20E increases (during the fast-feeding phase, days 5–7 of feeding), cellular $Ca^{2+}$ increases, mediating the cleavage of RhATG5, and thus the transition from autophagy to apoptosis.

autophagy in low concentrations (RhATG8-PE upregulated), while high concentrations of 20E stimulate the cleavage of RhATG5 (the appearance of NtRhATG5) and transition from autophagy to apoptosis (RhATG8-PE downregulated and cleaved-Rhcaspase7 upregulated) in tick salivary glands. However, the situation appears to differ in the salivary glands of blood sucking tick *in vivo*. Ticks switched from autophagy to apoptosis during the rapid blood-sucking period due to the increase of ecdysone in the hemolymph, but this transition was reversed post engorgement. Thus, our results indicated that the rates of both autophagy and apoptosis were high in tick salivary glands post engorgement. We hypothesize that this phenomenon might be due to the long feeding period of ticks, and might be caused by host-specific factors, but further studies are necessary to investigate these factors.

During the slow blood-feeding period, ecdysone concentration was elevated in the tick salivary glands, resulting in a general increase in the transcription of RhATGs. However, during the rapid feeding phase, ecdysone concentration fluctuated in the hemolymph, intracellular $Ca^{2+}$ concentration increased, and NtRhATG5 appeared (Fig 10). These factors reflected a transition from autophagy to apoptosis. Our findings provide a basis for advanced studies on tick salivary glands degeneration.

## Supporting information

**S1 Data. Excel spreadsheet containing, in separate sheets, the underlying numerical data and statistical analysis for Figure panels 1B, 2A, 2B, 3B, 3D, 6A, 6B, 7B, 7D, 9B, S2A and S2B Fig.**
(XLS)

**S2 Data. Word file containing all the orginal TIFFs for Figs 1A, 1C, 1D, 3A, 3C, 4A, 4B, 5A, 5B, 7A, 7C, 8A, 8B and 8C.**
(DOCX)

**S1 Fig. The RhATG5 protein.** (A) Alignment of the RhATG5 amino acid sequences with ATG5 protein sequences from *H. sapiens* (NP_004840.1), *M. musculus* (NP_444299.1), *H. armigera* (AMQ76404.1), and *D. melanogaster* (NM_132162.4). Conserved APG5 domain is underlined with a black line, and the critical conserved amino acid residue for ATG12-ATG5 interactions and complex formation are marked with a blue or black frame. Putative calpain cleavage sites of RhATG5 are marked with a red frame. The (*) indicates the position with single fully conserved residue, and (.) indicates conservation between groups with weakly similar properties. (B) Maximum likelihood phylogeny showing the relationship among RhATG5 (marked by a red dot) and the ATG5 homologs of other species. The evolutionary history was inferred using the Neighbor-Joining method [69]. The evolutionary distances were computed using the Poisson correction method [70] and are displayed in units of the number of amino acid substitutions per site.
(EPS)

**S2 Fig. Confirmation of RhATG5 silencing (*in vitro*) by qRT-PCR assays.** Transcription profiles of (A) fed-5d and (B) engorged-0d tick salivary glands after the RNAi of RhATG5. Bars represent mean relative expression of RhATG5 across three replicates; error bars represent standard error. ***, $P < 0.001$, based on two-tailed Student's *t* tests.
(TIF)

**S1 Table. Primers used for qRT-PCR.** S, forward primer; A, reverse primer.
(DOCX)

**S2 Table. Primers for apoptosis and autophagy related gene cloning and vector construction.** S, forward primer; A, reverse primer; Sm, forward primer for deletion mutation; Am, reverse primer for deletion mutation; the sequence in bold underlined indicates the sequence of the T7 promoter.
(DOCX)

## Acknowledgments

We thank LetPub (www.letpub.com) for its linguistic assistance during the preparation of this manuscript.

## Author Contributions

**Conceptualization:** Yanan Wang, Hiroshi Suzuki, Jinlin Zhou.

**Data curation:** Yanan Wang, Houshuang Zhang, Yongzhi Zhou.

**Formal analysis:** Yanan Wang, Houshuang Zhang, Xuenan Xuan, Hiroshi Suzuki.

**Funding acquisition:** Jinlin Zhou.

**Investigation:** Yanan Wang, Houshuang Zhang, Li Luo, Yongzhi Zhou, Jie Cao.

**Methodology:** Yanan Wang, Houshuang Zhang, Yongzhi Zhou, Xuenan Xuan.

**Project administration:** Houshuang Zhang, Jinlin Zhou.

**Resources:** Houshuang Zhang, Li Luo, Jinlin Zhou.

**Software:** Houshuang Zhang, Li Luo, Yongzhi Zhou.

**Supervision:** Jinlin Zhou.

**Validation:** Yanan Wang, Jinlin Zhou.

**Visualization:** Houshuang Zhang, Li Luo.

**Writing – original draft:** Yanan Wang.

**Writing – review & editing:** Yanan Wang, Xuenan Xuan, Hiroshi Suzuki, Jinlin Zhou.

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
