## [Decision Letter · Decision Letter 0]

13 Sep 2020

Dear Dr. Zhou,

Thank you very much for submitting your manuscript "ATG5 is instrumental in the transition from autophagy to apoptosis during the degeneration of tick salivary glands" for consideration at PLOS Neglected Tropical Diseases. As with all papers reviewed by the journal, your manuscript was reviewed by members of the editorial board and by several independent reviewers. In light of the reviews (below this email), we would like to invite the resubmission of a significantly-revised version that takes into account the reviewers' comments. 

We cannot make any decision about publication until we have seen the revised manuscript and your response to the reviewers' comments. Your revised manuscript is also likely to be sent to reviewers for further evaluation.

Sincerely,

Melissa J. Caimano

Deputy Editor

Reviewer's Responses to Questions

**Key Review Criteria Required for Acceptance?**

**Methods**

-Are the objectives of the study clearly articulated with a clear testable hypothesis stated?

-Is the study design appropriate to address the stated objectives?

-Is the population clearly described and appropriate for the hypothesis being tested?

-Is the sample size sufficient to ensure adequate power to address the hypothesis being tested?

-Were correct statistical analysis used to support conclusions?

-Are there concerns about ethical or regulatory requirements being met?

Reviewer #1: In the manuscript, Wang et al. investigated the roles of ATG5 in Rhipicephalus haemaphysaloides. The authors explored the role of ATG5 mediated the transition to apoptosis following 20E-induced autophagy, and most conclusions are well supported by the data. However, there are a few essential issues need to be addressed before it can be accepted for publication in the journal. 

Major concerns:

1. What's the different roles in ATG5 and ATG8 after 20E treated? The conclusion is not very clear.

2. The author must supply new clean WB of Fig3A, it's very hard to conclude from this WB plot, because they use polyclonal antibody, how they can distinguish the band of RhATG5/NtRhATG5 and RhATG8/PE? May need shRNA or KO data to support?

3. Figures need to be rearranged to 6-7 figs based on the result part.

4. Introduction part needs to be improved.

5. The authors must show more data of their RNAi experiments, it seems failed to RNAi from Fig8A.

Reviewer #2: (No Response)

**Results**

-Does the analysis presented match the analysis plan?

-Are the results clearly and completely presented?

-Are the figures (Tables, Images) of sufficient quality for clarity?

Reviewer #1: (No Response)

Reviewer #2: (No Response)

**Conclusions**

-Are the conclusions supported by the data presented?

-Are the limitations of analysis clearly described?

-Do the authors discuss how these data can be helpful to advance our understanding of the topic under study?

-Is public health relevance addressed?

Reviewer #1: (No Response)

Reviewer #2: (No Response)

**Editorial and Data Presentation Modifications?**

Reviewer #1: (No Response)

Reviewer #2: (No Response)

**Summary and General Comments**

Reviewer #1: (No Response)

Reviewer #2: This study by Wang et al., describes the importance of ATG5 in the transition from autophagy to aptosis during degeneration of tick salivary gland. This is a work that build previous knowledge about tick salary gland degeneration and is of great importance to better understand tick physiology. I have few comments, please see below.

Material and Methods:

Please describe with more details tick feeding, including number of ticks and rabbits used, time used for SG collection, etc.

Did authors perform an efficiency curve for primers used for qPCR?

Line 204: Does “fed” mean fully engorged ticks? Please clarify.

Salivary glands were cultivated and treated with 20E at µM concentration. What is the concentration of 20E in tick hemolymph? Would be informative to discuss this.

For RNAi and dsRNA synthesis, please describe the strategy used to avoid off-targets.

Results:

Figure 1: DAPI seems to be staining salivary gland cytoplasm at 3 day-fed.

Authors should include a BLAST analysis against the new 6 high-quality genomes described recently (https://www.cell.com/cell/pdf/S0092-8674(20)30931-4.pdf). Would be informative to have such comparison with different tick species regarding these salivary gland genes described here.

Please include statistical analysis for results described in Figure 2 (as done for Figure 6).

I would suggest authors to include as supplemental material full images of gels and WB.

Figure 4: Band intensity looks weak for a 10-20 µg load of calpain. Authors stated there is only one cleavage site for calpain and that RhATG5 was cleaved into one active fragment, however there are several bands (at least 3) resulted from this cleavage. 

Minor comments

Overall, the English is in good quality, but I would like to recommend authors to review manuscript carefully for typos.

Line 32: Please spell out ATG. There is no need to use capital letter for caspase (check all over the text).

Line 34: autophagy-related.

Line 177: isoelectric point

Line 225: (MKFQYKEEHPFEK).

Line 230: Missing a parenthesis after China.

Line 240: anti-alpha

Line 294: a total of

Figure S1: Please italicize species description

Line 333 and 334: autophagy and apoptosis.

PLOS authors have the option to publish the peer review history of their article (what does this mean?). If published, this will include your full peer review and any attached files.

Reviewer #1: No

Reviewer #2: No
---

## [Decision Letter · Decision Letter 1]

14 Dec 2020

Dear Dr. Zhou,

We are pleased to inform you that your manuscript 'ATG5 is instrumental in the transition from autophagy to apoptosis during the degeneration of tick salivary glands' has been provisionally accepted for publication in PLOS Neglected Tropical Diseases.

Best regards,

Melissa J. Caimano

Deputy Editor

Reviewer's Responses to Questions

**Key Review Criteria Required for Acceptance?**

**Methods**

-Are the objectives of the study clearly articulated with a clear testable hypothesis stated?

-Is the study design appropriate to address the stated objectives?

-Is the population clearly described and appropriate for the hypothesis being tested?

-Is the sample size sufficient to ensure adequate power to address the hypothesis being tested?

-Were correct statistical analysis used to support conclusions?

-Are there concerns about ethical or regulatory requirements being met?

Reviewer #1: (No Response)

Reviewer #2: (No Response)

**Results**

-Does the analysis presented match the analysis plan?

-Are the results clearly and completely presented?

-Are the figures (Tables, Images) of sufficient quality for clarity?

Reviewer #1: (No Response)

Reviewer #2: (No Response)

**Conclusions**

-Are the conclusions supported by the data presented?

-Are the limitations of analysis clearly described?

-Do the authors discuss how these data can be helpful to advance our understanding of the topic under study?

-Is public health relevance addressed?

Reviewer #1: (No Response)

Reviewer #2: (No Response)

**Editorial and Data Presentation Modifications?**

Reviewer #1: (No Response)

Reviewer #2: (No Response)

**Summary and General Comments**

Reviewer #1: (No Response)

Reviewer #2: (No Response)

PLOS authors have the option to publish the peer review history of their article (what does this mean?). If published, this will include your full peer review and any attached files.

Reviewer #1: No

Reviewer #2: No

---

## [Editor Report · Acceptance letter]

18 Jan 2021

Dear Dr. Zhou,

We are delighted to inform you that your manuscript, "ATG5 is instrumental in the transition from autophagy to apoptosis during the degeneration of tick salivary glands," has been formally accepted for publication in PLOS Neglected Tropical Diseases.

Best regards,

Shaden Kamhawi

co-Editor-in-Chief

Paul Brindley

co-Editor-in-Chief
